# Characterization of Natural Anthocyanin Indicator Based on Cellulose Bio-Composite Film for Monitoring the Freshness of Chicken Tenderloin

**DOI:** 10.3390/molecules27092752

**Published:** 2022-04-25

**Authors:** Athip Boonsiriwit, Pontree Itkor, Chanutwat Sirieawphikul, Youn Suk Lee

**Affiliations:** 1Department of Packaging, Yonsei University, Wonju 220-710, Korea; athip8266@gmail.com (A.B.); pontree.itkor@gmail.com (P.I.); 2Rattanakosin International College of Creative Entrepreneurship (RICE), Rajamangala University of Technology Rattanakosin, Nakhon Pathom 73170, Thailand; chanutwatgus@gmail.com

**Keywords:** anthocyanin, bio-composite film, hydroxypropyl methylcellulose, microcrystalline cellulose, pH sensing indicator, chicken tenderloin

## Abstract

Intelligent packaging with indicators that provide information about the quality of food products can inform the consumer regarding food safety and reduce food waste. A solid material for a pH-responsive indicator was developed from hydroxypropyl methylcellulose (HPMC) composited with microcrystalline cellulose (MCC). MCC at 5%, 10%, 20%, and 30% *w*/*w* was introduced into the HPMC matrix and the physical, barrier, thermal, and optical properties of the HPMC/MCC bio-composite (HMB) films were analyzed. At 5, 10, and 20% MCC, improved mechanical, transparency, and barrier properties were observed, where HMB with 20% of MCC (H20MB) showed the best performance. Therefore, H20MB was selected as the biodegradable solid material for fabricating Roselle anthocyanins (RA) pH sensing indicators. The performance of the RA-H20MB indicator was evaluated by monitoring its response to ammonia vapor and tracking freshness status of chicken tenderloin. The RA-H20MB showed a clear color change with respect to ammonia exposure and quality change of chicken tenderloin; the color changed from red to magenta, purple and green, respectively. These results indicated that RA-H20MB can be used as a biodegradable pH sensing indicator to determine food quality and freshness.

## 1. Introduction

Intelligent packaging can monitor quality changes in food products; therefore, it has garnered significant attention for application in food packaging. It can provide information on the direct quality of the food products within the package and indirectly inform the consumer of food safety, thereby protecting the consumer from food poisoning and reducing food wastage [1,2]. Based on the function of the detection indicators, intelligent packaging can be divided into three types: time–temperature, gas, and freshness indicators [3]. pH sensing indicators are one of the indicators for freshness, which change colors with respect to the pH of the environment [4]. Recently, a variety of colorimetric dyes from natural resources, such as anthocyanins, are increasingly being used in pH sensing indicator applications for monitoring food freshness [4,5,6]. Compared to the several synthetic dyes, anthocyanin presents a wide range in responding to the pH (Appendix A). The fabrication of anthocyanin-based pH sensing indicators require the incorporation of the anthocyanins in a solid material, for which biodegradable materials, such as chitosan, gelatin, and agar, are generally used [7]. 

However, these materials exhibit their own specific colors, which may directly affect the color of anthocyanin. Therefore, a colorless biomaterial is preferable for fabricating pH sensing indicators. As such, hydroxypropyl methylcellulose (HPMC), a biodegradable material, is a good candidate for fabricating pH sensing indicators because it consists of nonionic cellulose ethers, and it is colorless, odorless, tasteless, and flexible [8,9]. It does not interfere with the chemical structure and color of anthocyanin. Moreover, it is suitable for integration with a highly water-soluble substrate [8,10]. However, HPMC is transparent, has high water sensitivity, and exhibits inferior gas-barrier properties. Therefore, composite HPMC films with cellulose fibers have been recommended to improve these properties [11].

Microcrystalline cellulose (MCC), which is derived from high-quality wood pulp by removing the amorphous regions of cellulose by acid hydrolysis, can be a cellulosic reinforcement for polymers. MMC can also serve as a binder and disintegrating agent, facilitating film formation [8]. Several studies have reported that introducing MCC into polymers can improve their physical, optical, and barrier properties [12,13,14]. Therefore, this study aimed to develop biodegradable solid materials composed of HPMC with various concentrations of MCC. The physical, chemical, and barrier properties of the HPMC/MCC bio-composites (HMB) were analyzed. The optimum ratio of HMB was selected for fabricating an anthocyanin pH sensing indicator and its performance was evaluated by monitoring the color change after exposure to volatile ammonia and tracking the freshness status of chicken tenderloin. 

This study will provide a better understanding of cellulose bio-composite and intelligent packaging application with natural anthocyanin pH sensing indicator, where it is sensitive to the gases occurred by spoiled chicken tenderloin.

## 2. Results and Discussion

### 2.1. HMB Film Characterization

#### 2.1.1. Morphological Analysis

The morphology of the HPMC (control) and HMB films was observed by FE-SEM. The SEM images of the cross-section and surface of the films are shown in Figure 1. The cross-section of the HPMC film (Figure 1A) was smooth, where no stacking or cracking of the layers was observed. In contrast, introducing MCC into the HPMC structure (Figure 1B–D) showed a packed and homogenous layer between MCC and HPMC. Moreover, the surface images of HPMC (Figure 1F), H5MB (Figure 1G), H10MB (Figure 1H), and H20MB (Figure 1I) showed smooth areas without holes and agglomeration of MCC. These results indicated that 5–20% of the MCC content was successfully incorporated within the HPMC matrix. Several studies have previously reported that a polymer matrix can be improved by introducing an optimum concentration of MCC [12,14]. The high specific area and a large number of –OH groups in the MCC structure may facilitate the interaction with the polymer via hydrogen bonding [11,13]. However, further increasing the MCC content in the HPMC matrix significantly increased the stacked area and roughness of the surface cross-section, as observed in H30MB (30% MCC) (Figure 1E). The agglomeration effect is also particularly apparent for H30MB (Figure 1J). Excess MCC results in the disruption of the HPMC matrix, yielding a non-uniform film.

#### 2.1.2. Optical Properties 

The images and transparency values of the HPMC and HMB films are shown in Figure 2. A high transparency value denotes a less transparent film [15]. The transparency value of the HPMC film was 0.94 ± 0.31 Abs/mm. The introduction of MCC into the HPMC matrix increased the transparency values, with the transparency values of the H5MB, H10MB, H20MB, and H30MB were 1.96 ± 0.38, 2.81 ± 0.50, 5.12 ± 0.53, and 6.76 ± 0.45 Abs/mm, respectively. These results were consistent with the increase in the intensity of the white color of the HMB films (Figure 2A). This is in agreement with the results obtained by Hermawan, et al. [16], who reported that loading 7% of MCC into a seaweed film increased the transparency values up to 28.5%. 

#### 2.1.3. Physical Properties

The thickness of the composite film is an important factor because it directly affects the mechanical, barrier, and optical properties. The thicknesses of the control and composite films were between 73.62 ± 2.38 and 76.98 ± 3.57 μm (Table 1) with no significant differences.

Table 1 also lists the tensile strength and elongation at break values of the composite films. The tensile strength of HPMC was 24.08 ± 3.51 MPa. Introducing MCC into the HPMC matrix significantly increased the tensile strength of the HMB films, where the tensile strengths of the H10MB and H20MB films were 30.14 and 49.46% higher than that of HPMC, respectively. However, above 20% MCC, the tensile strength decreased, as observed in H30MB. As compared to H20MB, H30MB (a 10% increase in the MCC concentration) witnessed a 27.42% decrease in the tensile strength. These results indicate that 20% MCC was the optimum concentration for improving the mechanical properties of HPMC. The high specific area of the MCC fiber facilitates the formation of an adhesion net between the composite polymer and MCC during the film dehydration process, resulting in the improved tensile strength [17]. However, an excessive amount of MCC leads to a discontinuous microstructure, as observed in the SEM image, thereby decreasing the tensile strength.

The elongation at break of the HMB film gradually decreased compared to HPMC. This is frequently observed in bio-composite films that use cellulose fiber as a reinforcing element [14]. The elongation at break of HMB decreased from 33.19 ± 3.19 (control) to 28.17 ± 2.79, 25.23 ± 3.41, 24.96 ± 2.37, and 21.29 ± 3.36 for H5MB, H10MB, H20MB, and H30MB, respectively. This decreasing trend may be attributed to the stiffness of MCC, which directly hinders the mobility of the polymer strand [18]. These results were in agreement with the study conducted by Mathew, Oksman and Sain [12], who reported that introducing MCC into poly lactic acid (PLA) leads to a decrease in the elongation of the MCC/PLA composite films.

The gas barrier properties of the HMB films were determined by calculating the OTR and WVTR (Table 1). The OTR values of the control, H5MB, H10MB, H20MB, and H30MB were 51.97 ± 1.35, 20.37 ± 1.02, 14.00 ± 1.49, 6.85 ± 0.47, and 22.44 ± 0.96 cc^2^/m^2^/day, respectively, suggesting that MCC improved the gas barrier properties of HPMC. At 20% MCC, the film was least permeable to oxygen, which decreased the OTR value by 86.81% compared to the control, whereas above 20% MCC, the OTR value increased. This may be because at low concentrations, the MCC is well-dispersed within the HPMC matrix and acts as an effective barrier, slowing down the diffusion of gas molecules. However, increasing the MCC concentration leads to the formation of aggregates that generate voids within the HMB film, thereby facilitating the fast permeation of gases. These results were confirmed by SEM images of the cross-section and surface of the HMB film (Figure 1E,J).

The WVTR of the HPMC film was 48.17 g/m^2^/day. At 5 and 10% MCC, the moisture barrier properties did not improve, but at 20% MCC, the WVTR value reduced to 42.16 ± 3.21 g/m^2^/day (approximately by 12%), indicating an improvement in the moisture barrier properties. The presence of MCC in the polymer matrix increases the hydrophobicity of the composite films, thereby providing protection against moisture [19]. However, increasing the MCC content to 30% compromised the moisture barrier properties. This may be because at higher loadings, MCC aggregates formed voids within the film that allowed the fast permeation of gases, negating the barrier effect.

#### 2.1.4. XRD Analysis

XRD is a very useful technique for characterizing crystalline materials. XRD diffractograms of the composite films and CrI are shown in Figure 3. The main diffraction peaks of MCC were observed at 15° (110) and 22.5° (200), representing the crystalline polymorph I cellulose, which is the dominant form of cellulose in nature [20]. The CrI of MCC was 65.45%, which is within the range previously reported (65–83%) [21]. Increasing the MCC content leads to a significant increase in the diffraction peak at 22.5° and the CrI of the HMB films. The CrI of the HMB films ranged from 6.66 to 18.97% depending on the MCC concentration. As expected, the lowest and highest values of the CrI were exhibited by H5MB and H30MB, respectively. These results confirmed the presence of MCC, which was successfully incorporated into the HPMC matrix. Additionally, the CrI directly relates to the hydrophobicity of the material; a high value of CrI is indicative of higher hydrophobicity [22]. An increase in the CrI also relates to the brittleness of the composite film, which was confirmed by the results of the elongation at break (Table 1) as previously discussed. 

#### 2.1.5. Thermal Properties

The thermal degradation of the HMB films was determined by TGA in terms of percentage weight loss (Figure 4). All TGA curves (Figure 4A) of the HMB films showed a similar pattern to that of the HPMC film (control). The decomposition behavior of the HMB films observed in the DTG curves (Figure 4B) can be divided into two stages; the first stage begins at approximately 130 °C, which is attributed to the evaporation and degradation of glycerol in the composite films [16]; the second stage is observed at approximately 310 °C, representing the degradation of HPMC and MCC. 

Generally, the weight loss at 10% (T_10%_) and 50% (T_50%_) are used for comparing film stability [23]. There was no significant difference in the percent weight loss between the control and HMB films until T_10%_. However, an increase in the MCC content significantly decreased the degradation temperature of the composite films. The T_50%_ values of the control, H5MB, H10MB, H20MB, and H30MB films were 352.42 ± 1.45, 346.33 ± 1.15, 345.84 ± 1.26, 344.80 ± 0.99, and 337.83 ± 1.18 °C, respectively; there were no observed differences in the T_50%_ values of H5MB, H10MB, and H20MB. The decrease in thermal stability of the HMB may be due to the incompatibility and weak interfacial bonding between the polymer and fibers [24]. Similarly, Xian, Wang, Zhu, Guo, and Tian [14] reported a decrease in the thermal degradation temperature upon loading MCC in a PLA matrix. 

Based on the characterization results, the H20MB film showed the best physical and barrier properties and better optical properties than the control. Therefore, H20MB was selected for the subsequent fabrication of the RA anthocyanins pH sensing indicator. The performance of the RA-H20MB indicator was evaluated by monitoring the color change after exposure to volatile ammonia. In addition, the RA-H20MB indicator was applied in packaging for tracking the freshness status of chicken tenderloin.

### 2.2. RA and RA-H20MB Indicator Charaterization

#### 2.2.1. Color Response to pH of RA Anthocyanins

The color change of RA solution in the pH range of 1–12 is displayed in Figure 5A. RA solution exhibited a red shade which the intensity gradually decreased with the increase of pH at pH 1–4 and changed to light coral red at pH 5–6, magenta at pH 7–8, maroon at pH 9, grey at pH 10, brown at pH 11 and yellow at pH 12. The corresponding absorption spectra change of the RA in the pH range from 1 to 12 are shown in Figure 5B.

The maximum absorbance peak was obtained at approximately 518 nm of pH = 2; then, it gradually decreased and shifted to 579 nm when the pH was increased to 7. The change in absorbance peaks of RA solution is directly related to changes in the structures of anthocyanins [25]. This is because the molecular structure of anthocyanins has an ionic nature [26]. Generally, anthocyanins exist in the form of red flavylium cations at lower pH, which gradually change to colorless hemiketal and chalcones with an increase in pH, slowly transform into a blue quinoidal base under neutral and weakly alkaline conditions, and then turn to yellow chalcone under strongly alkaline conditions [27]. These results indicated that RA has the potential to be applied for the development of pH-sensing indicators due to it shows clear changes to a wide range of colors in response to variations in pH, which can be distinguished by the naked eye.

#### 2.2.2. RA-H20MB Indicator 

In order to evaluate the performance of the RA-H20MB indicator, NH_3_ at 0.8 M was employed as a representative of spoilage nitrogen compound in fresh meat. Figure 6 exhibits the change in color and ΔE value of H20MB and RA-H20MB indicator upon exposure to volatile NH_3_. Clearly, the color of RA-H20MB turned from red to magenta, purple, purple-blue, and green after 30 min of exposure. The ΔE values increased with exposure time which were 6.96–13.98 for magenta, 14.17–22.54 for purple, 35.54–36.80 for purple-blue, and 43.28–46.71 for green. The highest ΔE value of RA-H20MB was found at 28 min which value was 46.71 ± 1.90. In addition, the ΔE values of RA-H20MB were significantly higher than H20MB for all exposure times which confirmed that the change in color of the indicator originates from RA only. These results indicate that the RA-H20MB indicators were successfully developed and can be applied as freshness indicators for intelligent food packaging.

#### 2.2.3. Application of RA-H20MB Indicator for Tracking Freshness of Chicken Tenderloin

The freshness of chicken meat reduces with time, and the main factor for the reduction in freshness deals with the activity of various microorganisms [28]. The growth of microorganisms increases protein degradation which leads to an increase in the volatile nitrogen and pH of the packaging. According to this behavior, the freshness of chicken tenderloin can be tracked by monitoring the change of pH in the headspace gas. 

Figure 7 shows the parameters related to the freshness of chicken tenderloin and the ΔE value of the RA-H20MB. The initial TPC of chicken tenderloin was 3.90 ± 0.14 log CFU/g (Figure 7A), which gradually increased with the storage time to 8.20 ± 0.25 log CFU/g at 15 days of storage. According to, the TPC at 7 logs CFU/g was employed at the end of the microbiological shelf-life of fresh poultry [29]. Therefore, chicken tenderloin was marked as spoilage since 12 days of storage due to the TPC exceeded the acceptable limit. 

TVB-N at 40 mg/100 g has been used as the rejection limit for fresh chicken [30]. The TVB-N results of chicken tenderloin are presented in Figure 7A. The pattern of the TVB-N graph was similar to TPC results, increasing with the storage time. It increased from 8.43 ± 0.35 to 36.13 ± 0.35 mg/100 g at the end of storage period. However, there were no TVB-N results of chicken tenderloin that exceeded the rejection limit during the storage at 4 °C for 15 days.

The initial pH of chicken tenderloin was 6.13 ± 0.10 (Figure 7B), which increased with the storage time. At 15 days, the pH of chicken tenderloin reaches 7.00 ± 0.12. The higher pH at the end of storage is attributed to the accumulation of volatile compounds including ammonia, which originated from protein degradation owing to microbial activity [31]. 

The color change and ΔE values of RA-H20MB indicator are presented in Figure 7B. The RA-H20MB at the initial was red and turned to magenta at 6 days, red-wine at 9 days, purple at 12 days, and green at 15 days of storage period. The red color of RA-H20MB on the initial day was used as a reference to calculate ΔE values. The ΔE value increased with an increase in spoilage factors. It increased from 0 to 8.46 ± 1.6 on day 3, and gradually increased to 42.49 ± 1.80 at the end of the storage period. In addition, the empty PET tray (without chicken tenderloin) embedded with RA-H20MB was used as a control of the RA-H20MB indicator during the storage test. The result showed that there was no observed difference in the color change of the control indicator during 15 days of storage. Thus, these results indicated that the color of the RA-H20MB indicator changed according to the changes in the quality of chicken tenderloin. 

Images of chicken tenderloin packaging and changes in the color of the RA-H20MB indicator are presented in Figure 8. According to the changes in the color of the RA-H20MB indicator and spoilage results of chicken tenderloin including the standard of fresh chicken, each color of the RA-H20MB indicator could be interpreted as follows: red color demonstrated that chicken tenderloin was fresh; magenta and red wine demonstrated that chicken tenderloin was good to eat; purple and green revealed that chicken tenderloin was spoilage. These indicated that the RA-H20MB indicator can be used as a freshness indicator due to its performance and ability to change its color regarding the spoilage status of the chicken tenderloin sample. 

## 3. Materials and Methods

### 3.1. HMB Film Prepalation 

The HMB film was prepared according to a previously reported procedure [28] with some modifications. Briefly, the MCC (Sigma-Aldrich, St. Louis, MO, USA) suspension was prepared at 8% *w*/*w*. HPMC (Alfa Aesar, Haverhill, MA, USA) and was dissolved in DI water at 80 °C to obtain an HPMC solution at 5% *w*/*w*. Thereafter, the HPMC solution was mixed into the MCC suspension at different final ratios of HPMC: MCC = 100:0, 90:5, 90:10, 80:20, and 70:30 (dried basis), denoting HPMC (control), H5MB, H10MB, H20MB, and H30MB, respectively. Afterward, 6 g of glycerol was added to 400 mL of each HMB mixture. The mixture was heated at 80 °C, followed by homogenization using an IKA T-18 Ultra Turrax Digital Homogenizer (IKA, Wertheim, BW, Germany) at 10,000× *g* for 20 min. Subsequently, the temperature was reduced to 50 °C under the same homogenization conditions for 10 min, and the HMB solutions were degassed using a sonicator at 50 °C for 30 min. Finally, 30 mL of the HMB solutions were cast into a polystyrene petri dish (12 cm × 12 cm) and evaporated at 30 °C for 24 h to obtain HMB films. The dried films were wrapped with aluminum foil and stored in a desiccator at room temperature (25 °C) until further study.

### 3.2. Charactrization of HMB

#### 3.2.1. Field Emission Scanning Electron Microscopy (FE-SEM)

FE-SEM of the composite films was conducted using a Leica 360-S setup (Leica Microsystems Ltd., Cambridge, UK) at an acceleration voltage of 15 kV under high vacuum. A thin layer of Au was sputter-coated on the fractured surface of the composite films to avoid charging upon their exposure to an electron beam during SEM analysis.

#### 3.2.2. Mechanical Properties 

The thickness of each film was measured using a micrometer (Mitutoyo, Kawasaki, Japan). The tensile strengths and elongations of the films were tested using a universal texture meter according to GB 13022-1991. Before testing, the films were cut into rectangular strips of 6.0 cm × 2.0 cm and placed in the desiccator at room temperature (25 °C) for 24 h. The load cell and initial clamp distance were 150 kgf and 4.0 cm, respectively. The test speed was fixed at 0.06 mm/s. Each sample was analyzed five times at room temperature.

#### 3.2.3. Barrier Properties 

The oxygen transmission rate (OTR) of the films was evaluated with an 8001 Oxygen Permeation Analyzer (Illinois Instruments, Inc., Johnsburg, IL, USA). The oxygen gas permeability tests were performed at 23 °C and 0% relative humidity (RH) using high-purity nitrogen and oxygen as the carrier and testing gases, respectively. The water vapor transmission rate (WVTR) of the films was determined using a 7000 Water Vapor Permeation Analyzer (Illinois Instruments, Inc., Johnsburg, IL, USA). Measurements were performed at 38 °C and 90% RH.

#### 3.2.4. Optical Property

The transparency of the HMB films was measured according to the method proposed by Gutiérrez [32] at 600 nm and calculated using the following equation:(1)$$Transparency=\frac{A600}{e}$$
where *A*600 is the absorbance at 600 nm, and *e* is the film thickness (mm).

#### 3.2.5. X-ray Diffraction (XRD) Measurement

XRD tests were carried out using a Bruker AXS D8 Advance X-ray diffractometer (Bruker Inc., Ettlingen, Germany) accompanied by Ni-filtered Cu Kα radiation (λ = 0.154 nm) at a voltage of 40 kV and an electric current of 40 mA. The XRD profiles were obtained by scanning from 3 to 60° (2θ) at a rate of 3°/min. The crystallinity index (CrI) was calculated according to the Terinte, Ibbett and Schuster [21] method using the following Equation (2)
(2)$$CrI=\frac{I_{200}-I_{non-cr}}{I_{200}}\times 100$$
where *I*_200_ refers to the maximum intensity of the peak corresponding to the plane having a miller index of 200 (2θ = 22.5), while *I_non-cr_* represents the intensity of diffraction of the non-crystalline material.

#### 3.2.6. Thermal Properties

Thermogravimetric analysis (TGA) of the indicators was performed using a TGA 4000 system (PerkinElmer Co., Groningen, The Netherlands). TGA curves were obtained by heating the indicators from 30 to 800 °C at a rate of 10 °C/min under a N_2_ gas flow.

### 3.3. Application of HMB Film for Roselle Anthocyanin (RA) Indicator Fabrication

#### 3.3.1. Extraction and Characterization of RA 

Roselle (*Hibiscus sabdariffa*) was selected as a source of natural anthocyanins for fabricating the indicators due to the rich in effective anthocyanins and low cost. The freeze dried Roselle was purchased from a Happy Organic farm (Chiangmai, Thailand). RA was extracted following the method proposed by Zhang, Zou, Zhai, Huang, Jiang, and Holmes [25]. Briefly, 100 g of Roselle powder were added into 1500 mL of 80% ethanol, and pH of the mixture was adjusted to 2 using 1 M HCl. Then, the mixture was heated in a water bath at 50 °C for 1 h, filtered with Whatman No. 1 filter paper. Thereafter, the solvent was removed using a rotary evaporator (IKA Rotary evaporator RV 10 auto V, IKA, Staufen, Germany) at 50 °C in the dark. Finally, the RA extract was frozen and dried to obtain RA powder. The concentration of RA was determined according to the procedure demonstrated by Çam, et al. [33]. Briefly, RA solutions with pH = 1 and 4.5 was measured at 510 and 700 nm using a UV–Vis spectrophotometer (V-650 Spectrophotometer, JASCO Inc., Hachioji, Japan). The total anthocyanins content was determined as mg cyanidin/100 mL of RA using Equations (3) and (4):(3)$$A={\left(A_{510}-A_{700}\right)}_{pH=1}-{\left(A_{510}-A_{700}\right)}_{pH=4.5}$$
(4)$$total\;anthocyanin\;content=\frac{A\times M_{w}\times 100}{M_{A}}\;$$
where *M_W_* = molecular weight of cyanidin (449.2 Da) and *M_A_* = molar absorptivity of cyanidin (26,900).

The pH response of the RA was evaluated by changing the spectra. The RA powder was separately diluted with buffer solution (pH = 1–12) to obtain 3.75 mg of anthocyanins/100 mL of solution. Thereafter, the RA-HMB indicators at various pH values were scanned in the wavelength range of 400–800 nm using a UV–Vis spectrophotometer.

#### 3.3.2. Preparation of RA-HMB

To fabricate the RA-HMB indicator, 60 mg of RA was dissolved into 400 mL of HMB solution, homogenized at 5000× *g* for 5 min, and degassed for 30 min at 40 °C to obtain the RA-HMB solution. Thereafter, 30 mL of the RA-HMB solution was cast into a petri dish and dried at 30 °C for 24 h. 

#### 3.3.3. Color Response of RA-HMB to Ammonia 

The sensitivity of the indicators to volatile NH_3_ was determined based on a procedure formulated by Jiang, et al. [34] with some modifications. Briefly, the square (2 cm × 2 cm) RA-HMB indicators were placed in a 100 mL flask, 1 cm above an NH_3_ solution (60 mL, 0.8 M). Thereafter, at 2 min intervals, the changes in the colors of the indicators were recorded using a digital camera (Canon EOD 450D, Yokohama, Japan) and measured using a CR-400 Chroma Meter (Konica Minolta Sensing, Inc., Tokyo, Japan) to obtain *L**, *a**, and *b** values of the indicators at specific time intervals after exposure to NH_3_. The total color difference (Δ*E*) was calculated using Equation (5):(5)$$\Delta E=\sqrt{{\left(L^{*}-L_{0}^{*}\right)}^{2}+{\left(a^{*}\;-a_{0}^{*}\right)}^{2}+{\left(b^{*}-b_{0}^{*}\right)}^{2}}$$
where $L_{0,}^{*}$, $a_{0}^{*}$, and $b_{0}^{*}$ are initial color parameters of the indicators and *L**, *a**, and *b** are those at a given time.

#### 3.3.4. Application of RA-HMB Indicator for Tracking Freshness of Chicken Tenderloin

The performance of RA-HMB was evaluated by tracking the spoilage of chicken tenderloin which were purchased from local market in Wonju, Korea. For the packaging preparation, RA-HMB indicator (3 × 3 cm^2^) was attached under the cover of polyethylene terephthalate (PET) clamshell tray (250 mm × 150 mm × 90 mm). Thereafter, approximately 300 g of fresh chicken tenderloins were placed in prepared tray. Each packed tray was hot-sealed (ISS 350–5, Gasungpak Co., Busan, South Korea) in a linear low-density polyethylene pouch (thickness = 60 μm) and stored at 4 °C in the dark for 15 days. 

Samples were subjected to be analyzed in 3 day intervals. The images and the color parameters of the RA-HMB indicator were determined according to a previously mentioned method. The spoilage parameter, total bacterial count (TPC), and total volatile basic nitrogen (TVB-N) of chicken tenderloin were performed at room temperature according to method suggested by Baltić, Ćirić, Velebit, Petronijević, Lakićević, Đorđević and Janković [30]. The results of TPC and TVB-N were expressed as log_10_ CFU/g and mg/100 g of chicken meat, respectively. The pH values of the samples were evaluated by a digital pH meter (AB15pH, Fisher Scientific Co., Pittsburgh, PA, USA) after mixing 10 g of the chicken tenderloin in 90 mL of distilled water.

### 3.4. Statistical Analysis 

All experiments were conducted at least in triplicates. The experimental data were analyzed using Statistical Package for the Social Sciences (SPSS) (SPSS 25 for Windows, SPSS Inc., Chicago, IL, USA) and analysis of variance (ANOVA). Statistical significance of the difference in the mean values was established by *p* ≤ 0.05, and Duncan’s new multiple range test was utilized for all statistical analyses.

## 4. Conclusions

A biodegradable solid material by compositing HPMC and MCC was successfully developed. Introducing 5%, 10%, and 20% of MCC into the HPMC matrix improved the physical, barrier, and optical properties of HPMC. H20MB exhibited the best physical and barrier properties and better optical properties as compared to the control. Therefore, H20MC was selected for fabricating the RA pH sensing indicator. The RA-H20MB indicator exhibited a clear change in color in response to various pH conditions and ammonia vapor exposure times. Moreover, the RA-H20MB was applied to track the freshness of chicken tenderloin. It was found that the color change of RA-H20MB was easy to distinguish and related to the change in quality parameters of chicken tenderloin. These results indicated that the novel anthocyanin pH sensing indicator synthesized in this study could be a potential candidate for the environment-friendly pH sensing indicators, which can provide consumers with valuable information regarding food quality.

## Figures and Tables

**Figure 1 molecules-27-02752-f001:**
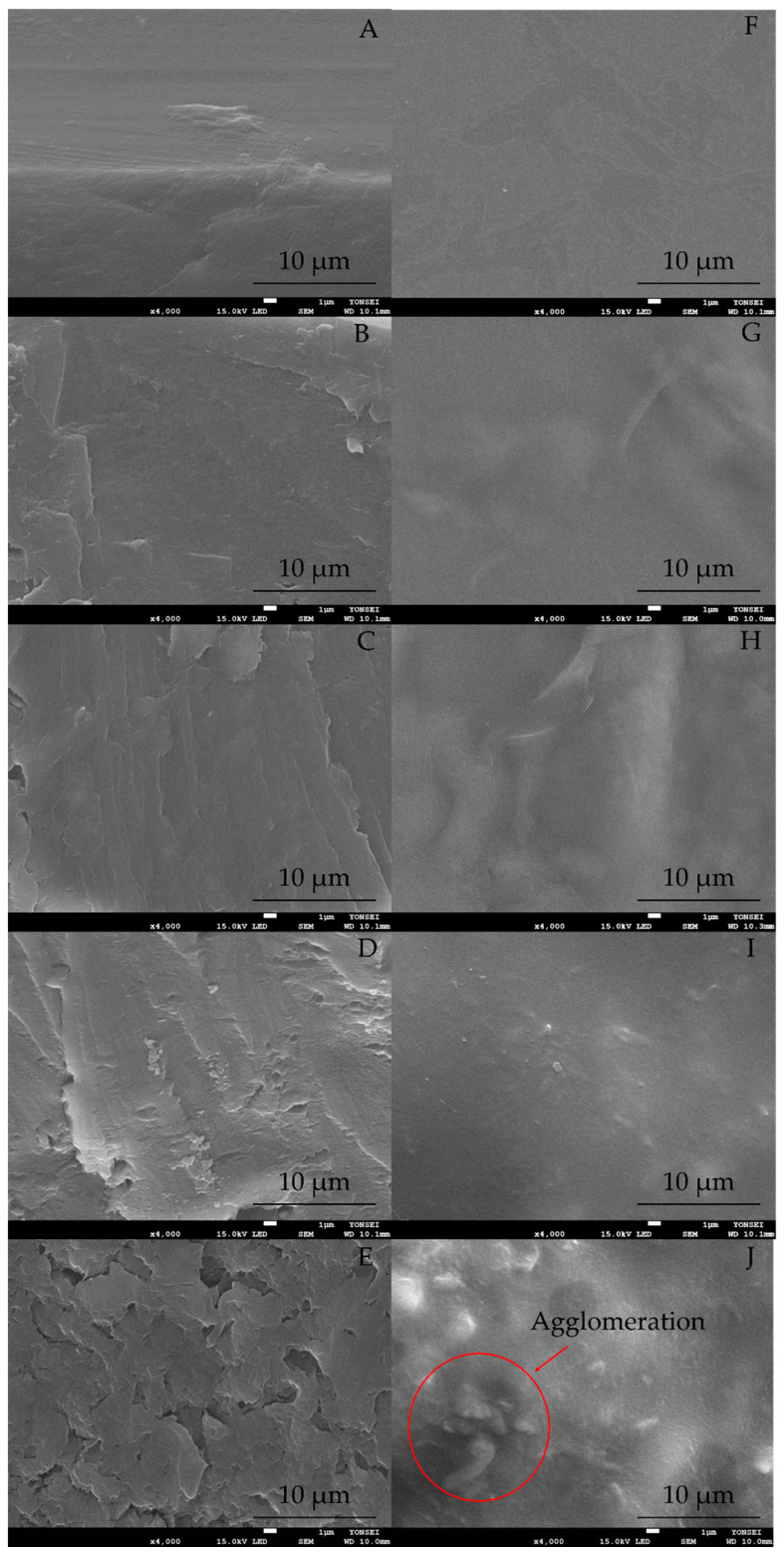
Field emission scanning electron microscopy images of the cross-section of the pure hydroxypropyl methylcellulose (HPMC) (**A**), 5% microcrystalline cellulose (MCC) (H5MB) (**B**), 10% MCC (H10MB) (**C**), 20% MCC (H20MB) (**D**), and 30% MCC (H30MB) (**E**) films and the surface of the HPMC (**F**), H5MB (**G**), H10MB (**H**), H20MB (**I**), and H30MB (**J**) films.

**Figure 2 molecules-27-02752-f002:**
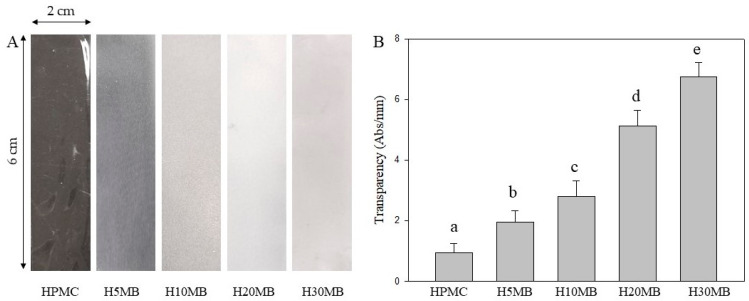
Image (**A**) and transparency values (**B**) of the hydroxypropyl methylcellulose (HPMC), 5% microcrystalline cellulose (MCC) (H5MB), 10% MCC (H10MB), 20% MCC (H20MB), and 30% MCC (H30MB) films. a–e letters indicate significant differences (*p* < 0.05).

**Figure 3 molecules-27-02752-f003:**
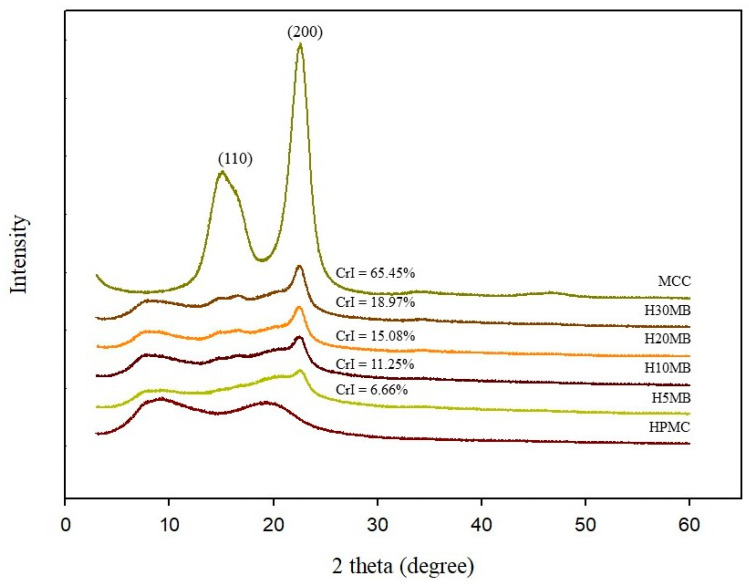
X-ray diffraction spectra of the hydroxypropyl methylcellulose (HPMC), 5% microcrystalline cellulose (MCC) (H5MB), 10% MCC (H10MB), 20% MCC (H20MB), and 30% MCC (H30MB) films.

**Figure 4 molecules-27-02752-f004:**
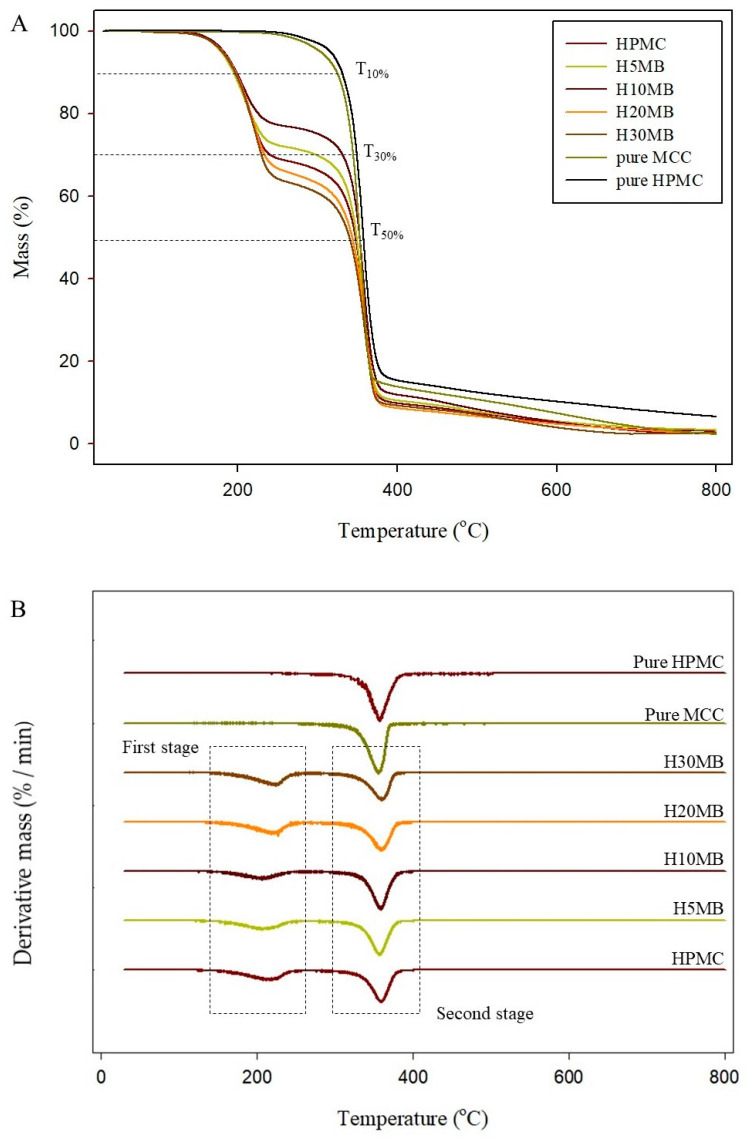
Thermogravimetry (**A**) and differential thermogravimetry (**B**) curves of the hydroxypropyl methylcellulose (HPMC), 5% microcrystalline cellulose (MCC) (H5MB), 10% MCC (H10MB), 20% MCC (H20MB), and 30% MCC (H30MB) films.

**Figure 5 molecules-27-02752-f005:**
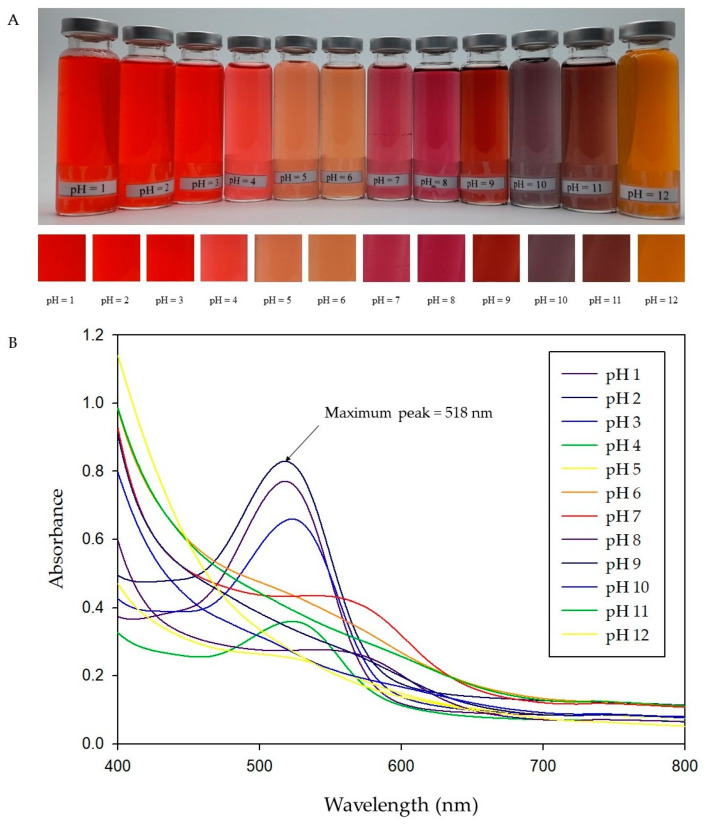
Changes in colors of RA solution (**A**) and UV–vis spectra of RA solutions in the pH range of 1–12 (**B**).

**Figure 6 molecules-27-02752-f006:**
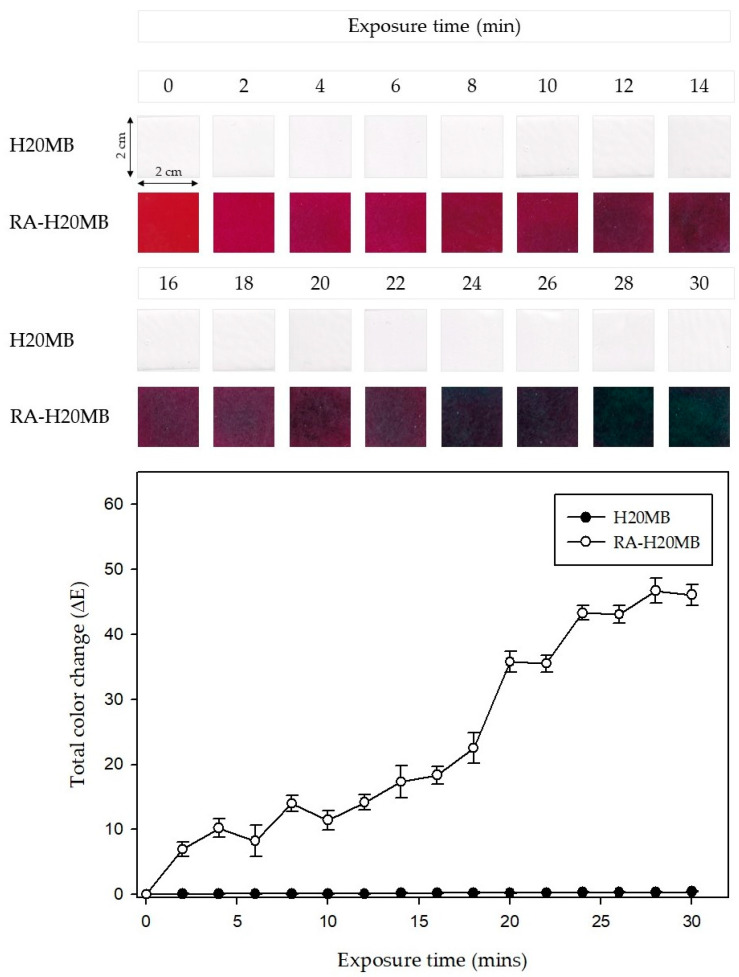
Total color change (ΔE) of the RA-H20MB indicators upon exposure to NH_3_ vapors.

**Figure 7 molecules-27-02752-f007:**
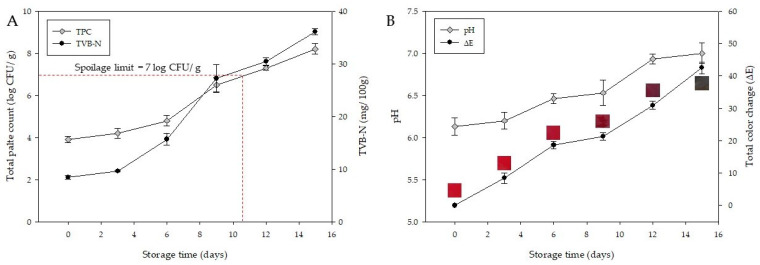
The changes of TPC and TVB-N of chicken tenderloin (**A**) and the change of pH of chicken tenderloin and the total color change (ΔE) of the RA-H20MB indicator (**B**) during stored at 4 °C.

**Figure 8 molecules-27-02752-f008:**
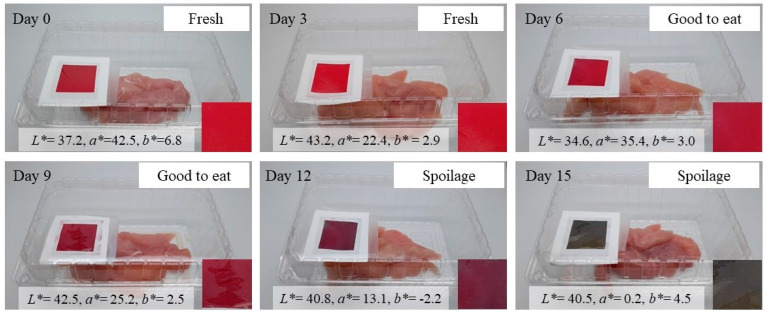
The color change of the RA-H20MB indicator and freshness/spoilage status of chicken tenderloin stored at 4 °C.

**Table 1 molecules-27-02752-t001:** Characteristics of the hydroxypropyl methylcellulose (HPMC), 5% microcrystalline cellulose (MCC) (H5MB), 10% MCC (H10MB), 20% MCC (H20MB), and 30% MCC (H30MB) films.

Sample	Thickness (μm)	Tensile (MPa)	Elongation (%)	OTR (cc^2^/m^2^/day)	WVTR (g/m^2^/day)
HPMC	74.59 ± 1.48 ^ns^	24.08 ± 3.51 ^a^	33.19 ± 3.19 ^c^	51.97 ± 1.35 ^d^	48.17 ± 3.97 ^bc^
H5MB	73.62 ± 2.38 ^ns^	28.17 ± 3.73 ^ab^	28.17 ± 2.79 ^bc^	20.37 ± 1.02 ^c^	45.00 ± 1.37 ^ab^
H10MB	75.84 ± 3.64 ^ns^	31.34 ± 2.86 ^bc^	25.23 ± 3.41 ^ab^	14.00 ± 1.49 ^b^	43.31 ± 2.19 ^ab^
H20MB	74.48 ± 3.48 ^ns^	35.99 ± 1.88 ^c^	24.96 ± 2.37 ^ab^	6.85 ± 0.47 ^a^	42.16 ± 3.21 ^a^
H30MB	76.98 ± 3.57 ^ns^	26.12 ± 2.84 ^ab^	21.29 ± 3.36 ^a^	22.44 ± 0.96 ^c^	51.93 ± 1.33 ^c^

Data represented means ± SD (*n* = 5), ^ns^ means not significantly different, ^a–d^ means with same superscript in a column do not vary significantly (*p* < 0.05) with respect to each other.

## Data Availability

The data supporting the findings of this study are included in this article.

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
