# Peer review of "Characterization of Natural Anthocyanin Indicator Based on Cellulose Bio-Composite Film for Monitoring the Freshness of Chicken Tenderloin"

_molecules, 2022, doi:10.3390/molecules27092752_

Round 1
Reviewer 1 Report
In my opinion, this a very well written paper and scientifically solid. It may be accepted for publication after the following minor revisions:
- Fig. 4A, according to ICTAC recomendations, please replace 'weight [%]' with 'mass [%]' on the y axis/
- Fig. 5B, is it correct wavenumber on x axys? Shoudn't it be λ [nm]?
- Auhtors only differences in colour parameters (i.e. ΔE). I think that the individual colour parameters (L, a*,b*) should also be given.
Author Response
Dear reviewers,
I've corrected the revision manuscript according to all reviewer's comments.
Please see the attachment file.
Thank you for your time.

Reviewer 2 Report
The paper "Manufacture and characterization of pH detection indicator of natural anthocyanin extract based on hydroxypropyl methylcellulose/microcrystalline cellulose biocomposite and its application to track chicken loin freshness "It's very important, elaborated and brings results. I observed that the references are current and that the results are consistent with the literature and with the proposed methodology. Nonetheless, I have some doubts. 1. line 387 - a total bacterial count was made. At what temperature was this assessment performed? the storage was done at 4ºC, and the counting was also done? 2. How about a control test, with the empty packaging and the sensor, to verify if the change in placement also happens due to time and photooxidation? Consider this control important.
Author Response

(The authors gave the same response as above.)

Reviewer 3 Report
Manuscript molecules-1698903 is well written and deserves publication.
Some suggestion to improve the manuscript:
- Paper titles are usually 100-150 characters long, with spaces included, if possible paper title could be shortened.
- At Introduction or in the supplementary material give a table with a comparison with other pH sensing indicators for food freshness tracking.
- At the end of the Introduction emphasize the novelty of the paper.
- Line 71: correct (1F) throughout
- Increase text size (legibility) of bottom bar SEM micrographs
- Fig 2A and Fig 6 : insert color and size scale bars in the films image
- Line 152: 2.1.3. should be 2.1.4.
- Line 171: 2.1.4. should be 2.1.5.
- 4B indicate y scale name. Indicate exo/endo direction
- Fig.5 indicate peak maxima on the spectra or in the legend bar
- Unite Fig 7A and 7B in one Figure, displaying the y axes to right and to the left, with different colors for an easier analysis. The same for Fig 7C and 7D.
- Line 343: power should be powder.
Author Response

(The authors gave the same response as above.)
